# Nocebo-Prone Behavior Associated with SARS-CoV-2 Vaccine Hesitancy in Healthcare Workers

**DOI:** 10.3390/vaccines9101179

**Published:** 2021-10-14

**Authors:** Dimos D. Mitsikostas, Konstantina Aravantinou-Fatorou, Christina Deligianni, Evrydiki Kravvariti, Eleni Korompoki, Maria Mylona, Pinelopi Vryttia, Georgia Papagiannopoulou, Eumorphia-Maria Delicha, Athanasios Dellis, Georgios Tsivgoulis, Meletios A. Dimopoulos, Martina Amanzio, Petros P. Sfikakis

**Affiliations:** 11st Neurology Department, Aeginition Hospital, Medical School, National and Kapodistrian University of Athens, 11528 Athens, Greece; k.h.aravantinou@gmail.com; 2Danish Headache Center, Rigshospitalet Glostrup, University of Copenhagen, 2600 Copenhagen, Denmark; cdchristina@gmail.com; 31st Department of Propaedeutic Internal Medicine, Joint Rheumatology Program, Laikon General Hospital, Medical School, National and Kapodistrian University of Athens, 11527 Athens, Greece; evkrav@med.uoa.gr (E.K.); mamylona@med.uoa.gr (M.M.); psfikakis@med.uoa.gr (P.P.S.); 4Department of Clinical Therapeutics, Alexandra Hospital, Medical School, National and Kapodistrian University of Athens, 11528 Athens, Greece; e.korompoki@imperial.ac.uk (E.K.); pinelopivrt@gmail.com (P.V.); mdimop@med.uoa.gr (M.A.D.); 52nd Neurology Department, Attikon Hospital, Medical School, National and Kapodistrian University of Athens, 12462 Athens, Greece; georgiapap22@hotmail.com (G.P.); tsivgoulisgiorg@yahoo.gr (G.T.); 6ASTAT, Statistics in Clinical Research, 16675 Glyfada, Greece; edelicha@astat.gr; 72nd Surgery Department, Aretaieion Hospital, Medical School, National and Kapodistrian University of Athens, 11528 Athens, Greece; aedellis@gmail.com; 8Department of Psychology, University of Turin, Via Verdi 10, 10124 Turin, Italy; martina.amanzio@unito.it

**Keywords:** SARS-CoV-2, nocebo, vaccine hesitancy, healthcare workers, tolerability, adverse event

## Abstract

Among healthcare workers (HCWs), SARS-CoV-2 vaccine hesitancy may be linked to a higher susceptibility to nocebo effects, i.e., adverse events (AEs) experienced after medical treatments due to negative expectations. To investigate this hypothesis a cross-sectional survey was performed with a self-completed questionnaire that included a tool (Q-No) for the identification of nocebo-prone individuals. A total of 1309 HCWs (67.2% women; 43.4% physicians; 28.4% nurses; 11.5% administrative staff; 16.6% other personnel) completed the questionnaires, among whom 237 (18.1%) had declined vaccination. Q-No scores were ≥15 in 325 participants (24.8%) suggesting nocebo-prone behavior. In a multivariate logistic regression model with Q-No score, age, gender, and occupation as independent variables, estimated odds ratios (ORs) of vaccination were 0.43 (i.e., less likely, *p* < 0.001) in participants with Q-No score ≥ 15 vs. Q-No score < 15, 0.58 in females vs. males (*p* = 0.013), and 4.7 (i.e., more likely) in physicians vs. other HCWs (*p* < 0.001), independent of age, which was not significantly associated with OR of vaccination. At least one adverse effect (AE) was reported by 67.5% of vaccinees, mostly local pain and flu-like symptoms. In a multivariate logistic regression model, with Q-No score, age, gender, and occupation as independent variables, estimated ORs of AE reporting were 2.0 in females vs. males (*p* < 0.001) and 1.47 in physicians vs. other HCWs (*p* = 0.017) independently of age and Q-No score, which were not significantly associated with OR of AE. These findings suggest that nocebo-prone behavior in HCWs is associated with SARS-CoV-2 vaccination hesitancy indicating a potential benefit of a campaign focused on nocebo-prone people.

## 1. Introduction

COVID-19 continues to spread worldwide, and acceptance of anti- SARS-CoV-2 vaccines is crucial for achieving sufficient immunization to impede the pandemic [1]. Healthcare workers (HCWs) represent a priority group for vaccination because of their elevated risk of SARS-CoV-2 infection and potential contribution to nosocomial transmission. However, evidence suggests that a significant proportion of HCWs are hesitant toward SARS-CoV-2 vaccination, inadvertently lending support to the anti-vaccination movement [2,3,4], along with misinformation and disinformation—the so-called “infodemic” [5].

The behavioral decision to accept, delay, or reject some or all vaccines have been classified by the Vaccine Hesitancy Determinants Matrix under three categories: contextual, individual and group, and vaccine/vaccination-specific influences [6,7]. Strategies implemented so far in order to address vaccine hesitancy have included multi-component interventions focused on raising knowledge and awareness [8,9,10], but the nocebo effect has not been investigated or acknowledged as a potential contributor. The nocebo effect, and its opposite, the placebo effect, refer to health changes attributed to negative and positive patients’ expectations, respectively, rather than pharmacological or physical implications of the treatment [11]. Although both nocebo and placebo effects are very powerful and pervasive in clinical practice, they are underestimated and underappreciated [12].

Placebo effects benefit medical treatment, yet nocebo effects limit it considerably, undermining treatment tolerability and adherence, and leading to treatment withdrawal [13]. In double-blind clinical trials testing treatments for neurologic or rheumatologic diseases, one out of 20 to one out of 10 patients in the placebo group discontinue treatment because of adverse events (AEs), although they are not treated with actual medication; a proportion of these AE reports correspond to nocebo effects, while others are due to spontaneous disease fluctuation or random comorbidities [14,15,16]. In a decade-long nationwide study of post-vaccination AEs in France, non-live virus vaccine recipients preferentially reported AEs associated with the disease that those vaccines were meant to prevent, for example, trismus for the tetanus vaccine or gynecological symptoms for the HPV vaccine [17].

Factors known to influence nocebo effects include personality traits, previous medical experiences, social observational learning, negative conditioning, comorbid anxiety and other mood disorders, and the physician’s communication strategies. Media or internet information is also essential in modifying placebo or nocebo effects [11,13]. Thus, the plethora of negative media information for the SARS-CoV-2 vaccines delivered worldwide daily may generate nocebo behaviors leading to vaccine hesitancy [5,18], which is identified by WHO as a delay in acceptance or refusal of vaccination despite availability of vaccination services (Box 1) [6].

Box 1Putting research in context.
**Evidence before this study**
The nocebo effect refers to unfavorable health changes after medical treatment that are attributable to patients’ negative expectations and not the treatment itself, and it has been associated with increased reporting of vaccine-related adverse effects. Despite the crucial significance of SARS-Cov2 vaccination in the global battle against the COVID-19 pandemic, there is a non-negligible proportion of healthcare workers (HCWs) that experience exaggerated fears of SARS-CoV-2 vaccine-related AEs and hesitancy, which could be linked to nocebo-related mechanisms.
**Added value of this study**
This study evaluated the rates of predisposition to nocebo effects among HCWs, along with attitudes toward vaccination and reported AEs. The main findings are that nocebo-prone behavior is associated with reluctance towards SARS-CoV-2 vaccination but not with post-vaccination AE reporting.
**Implications of all available evidence**
SARS-Cov-2 vaccine hesitancy and nocebo-prone behavior may share common underlying mechanisms, therefore, communication strategies targeting the nocebo are worthy of investigation as a potential mitigation.

In clinical practice, the frequency and intensity of nocebo effects is difficult to determine. Among questionnaires capturing beliefs about medications, the Q-No tool, a four-item questionnaire screening for behaviors that indicate high levels of negative expectations and fear regarding potential drug-related AEs, can predict a neurologist’s opinion that a person is prone to nocebo effects [19]. The aim of the present study is to investigate the hypothesis that a predisposition to the nocebo effect, as captured by the Q-No tool, is associated with SARS-CoV-2 vaccination hesitancy and/or increased reporting of post-vaccination AEs among HCWs.

## 2. Methods

### 2.1. Study Design, Participants, and Instruments Used

A cross-sectional survey was carried out from 1 May to 31 July 2021 to investigate attitudes towards vaccination and reported AEs following the SARS-CoV-2 vaccine among HCWs in five tertiary hospitals in Athens, Greece (Figure 1). At the time of the study, all HCWs employed at all five hospitals had been offered the option and were encouraged to vaccinate (the Greek SARS-Cov-2 HCW vaccination program has operated since 1 January 2021). The study used a self-completed questionnaire with 14 close-ended questions about demographics, the type of the vaccine they received or not, as well as the reasons why they made the decision to be vaccinated or not to be vaccinated. The questionnaire also included a section related to post-vaccination AEs according to the HCWs own evaluation, and if they required medical attention. Finally, the four-item Q-No questionnaire was included, a questionnaire with total score range of 4–20 points (Table 1). Using a cut-off at score 15 the Q-No has 71.7% specificity and 67.5% sensitivity for presumed nocebo susceptibility.

All HCWs (e.g., physicians, nurses, paramedical workers, and administrative and housekeeping staff), from representative departments of five tertiary hospitals (official staff) in Athens (Aeginition, Aretaieion, Alexandra, Laikon, and Attikon Hospital), were considered for participation. In each hospital, a separate research team was recruited, and team members handed out the questionnaires to all staff members on duty at the selected departments, so as to obtain consent forms, explain the study aims, and deliver detailed instructions. Participants were then left to complete questionnaires and place them in a sealed return envelope, to preserve their anonymity and the de-identified nature of the data. The study protocol, which is in accordance with the declaration of Helsinki, was approved by the Ethic Committees of all five hospitals (Figure 1).

### 2.2. Statistics

Categorical variables were described based on absolute and relative frequencies, whereas for continuous ones, mean and standard deviation (SD) were provided. The 95% confidence intervals for the corresponding proportions were calculated based on the logit transformation. Univariate comparisons were based on Pearson’s Chi-square test without continuity correction or Mc Nemar’s chi squared statistic as appropriate. For the age comparison between nocebo categories, *t*-tests were used. Multivariable logistic regression was employed to investigate the association between the Q-No score and SARS-CoV-2 vaccination hesitancy, adjusted for gender, age, and occupation. Sensitivity analysis on vaccination was also performed by multivariate logistic regression excluding subjects with relative or absolute contraindications to vaccination (such as known allergies, etc.). Statistical analysis was conducted in STATA 15.1 by E-M.D. and the significance level was set to α = 0.05.

## 3. Findings

During the pre-specified study period, 1543 individuals from the personnel of the five hospitals were approached, interviewed, read the study protocol, and invited to participate; 1400 chose to participate in the study (90.7%). After exclusion of 91 questionnaires due to inconsistent responses, 1309 questionnaires were analyzed. With regard to gender, 421 (32.6%) were males and 866 (67.2%) were females; three (0.16%) self-identified as non-binary. Basic epidemiological characteristics by gender, occupation, SARS-CoV-2 vaccination status, and Q-No score are presented in Table 2.

### 3.1. Vaccination Status and Hesitancy

Out of 1309 participants, 1072 (81.9%) received the first dose and 1042 (79.6%) were fully vaccinated, either with two doses or by a single-dose vaccine in seven cases (Table 2). Most vaccinated participants (95.9%) underwent the Pfizer/BNT162b2 vaccine because this was accessible at their hospitals during the survey. Among 30 participants who had not received a second dose, this had been planned and pending; three participants avoided the second dose because they experienced AEs after the first dose, and eight reported that they did not perform the second dose because of breakthrough infection after the first dose. Regarding their motivation, most vaccinated participants (948/1042) responded that they opted to vaccinate because they believed it was beneficial to themselves and society in general; 68/1042 felt obligated to do so in their professional duty; and 25/1042 gave other reasons in free text (e.g., “I’m afraid of getting sick and dying”, “I’m afraid of sticking my elderly parents”, etc.).

Out of 1309 participants in total, 237 (18.1%, 95%: 16.1–20.3) opted against vaccination (Table 2), due to: (i) fears related to vaccine safety (100/237); (ii) doubts regarding vaccine efficacy (28/237); (iii) skepticism regarding the need for mass vaccination (8/237); and (iv) other reasons (115/237) (participants could provide more than one answer).

### 3.2. Nocebo-Prone Behavior and Vaccine Hesitancy

In the Q-No questionnaire, 325 participants evenly distributed among the five participating hospitals (24.8%; mean age ± SD 43.3 ± 11.4) scored ≥ 15, suggesting nocebo-prone behavior. The remaining 984 participants (75.2%, 95% CI: 72.8%–77.4%; mean age ± SD 40.9 ± 10.7) scored less than 15 in Q-No; they were more commonly physicians rather than other HCWs, and men rather than women (*p*-value < 0.001 and *p* = 0.003, respectively). On average, participants with Q-No score ≥ 15 were 2.36 years older than participants with Q-No score < 15 (*p* = 0.0015, Table 2).

In a multivariate logistic regression model (Table 3), with logit of vaccine acceptance as the dependent variable (the outcome), and Q-No score, age, gender, and occupation as the independent variables (hypothesized predictors), estimated odds ratios (ORs) of vaccination were 0.43 (i.e., vaccination less likely) in participants with Q-No score ≥ 15 compared to those with Q-No score < 15 (*p* < 0.001 for the null hypothesis), 0.58 (vaccination less likely) in females compared to males (*p* = 0.013), and 4.7 (vaccination more likely) in physicians compared to other HCWs (*p* < 0.001), independent of age, which was not significantly associated with OR of vaccination.

### 3.3. Adverse Events Reported after Vaccination

At least one AE was reported by 67.5% of participants after complete vaccination (either after both doses or the single-shot vaccine). After the first dose of vaccine 50.1% of recipients reported any AE and 56.2% after the second dose; 5.9% more vaccine recipients reported an AE after the second vaccine dose (*p*-value = 0.001). Most common AEs included local pain, headache, fever, myalgia, and fatigue (Table 4, Table 5 and Table 6). Among vaccinee participants, 207 (40.3%) and 308 (54.2%) received medical treatment (most often over-the-counter analgesics) to manage the AEs they had experienced after the 1st or 2nd dose received, respectively.

Multivariate logistic regression analysis for reporting AEs among vaccinees, including as covariates the Q-No score, gender, age, and occupation (Table 7), revealed that female vaccinees (compared to males) and physician (compared to other HCWs) were more likely to report vaccination-related AEs.

## 4. Discussion

### 4.1. Hesitancy to Anti-SAR-CoV-2 Vaccines and Nocebo-Prone Behavior among HCWs

This cross-sectional, multicenter, face-to-face survey among HCWs revealed that 18.1% of participants decided not to receive the SARS-CoV-2 vaccine, although the vaccination program was already in operation for 4 months, thus fulfilling the criteria for vaccine hesitancy [6]. Regression analysis revealed that participants with a Q-No score ≥ 15 suggesting nocebo-prone behavior, as well as females, had higher probability for hesitancy. In our country, a previous cross-sectional study to investigate the intention for anti-SARS-CoV-2 vaccination completed in December 2020 prior to the start of the vaccination program, revealed that the proportion of hesitant Greek HCWs was 21.5%, congruent with our results [23]. However, the association between vaccine hesitancy and a predisposition to nocebo effects has not been investigated previously.

Our findings provide evidence that nocebo prone behavior and by extension, the nocebo effect, is linked to vaccine hesitancy, a crucial challenge amid the COVID-19 pandemic: In two French cross-sectional studies investigating the intention for vaccination, hesitancy towards theSARS-CoV-2 vaccine reached 25–27% among HCWs [4,24]. Similar to our findings, non-physicians (e.g., paramedical and administrative staff) were less likely to accept vaccination than physicians [4,24]. Among German HCWs the intention to get a vaccine for SARS-CoV-2 infection was as high as 91.7% (4125/4500) [2], but in the UK 23% of HCWs reported vaccine hesitancy in a large-scale web-based survey [3]. In the latter study, HCWs with non-European origin were more likely to be vaccine hesitant, while in other studies in low- and middle-income countries outside Europe, vaccine hesitancy has been found lower than in the US and Russia [1]; these findings indicate that cultural and social factors are also involved in the development of hesitancy among HCWs.

In our study and in other reports [3,4,12,25] female participants were more likely to report hesitancy and/or AEs from SARS-CoV2 vaccination, which may reflect additional concerns regarding reproductive adverse effects, but may also be related to a higher likelihood of nocebo-prone behavior in women, since previous research has identified female gender as a risk factor for nocebo effects in general [26,27,28,29].

As in our study, other surveys in Europe also report that overall, non-physicians were less prone to accept vaccination than physicians [10]. A significantly lower proportion of physicians scored more than 15 in Q-No, and this association may also explain why physicians experience vaccine hesitancy less often than other HCW occupations. Several other factors may contribute to vaccine hesitancy—cultural, financial, social, idiosyncratic, and religious dynamics have been associated with attitudes toward vaccination, and the same factors place patients at risk for nocebo effects [11,29]. The role of age in vaccine hesitancy was not very important in our study population.

### 4.2. Adverse Events in Anti-SAR-CoV-2 Vaccines and Nocebo Effect among HCWs

In our study, 67.5% of vaccinees reported AEs after at least one vaccination dose. In the pivotal study, the most common AEs of BNT162b2 mRNA COVID-19 vaccine were mild-to-moderate pain at the injection site, fatigue, and headache. As in our study, only a few recipients used medical treatments for these AEs, and systemic AEs were more prevalent after the second dose [30]. HCWs in the Czech Republic reported at least one post-vaccine AE in the vast majority of vaccine recipients (93.1%), including injection site pain (89.8%), followed by fatigue (62.2%), headache (45.6%), muscle pain (37.1%), and chills (33.9%) [25]. False safety signals for vaccines have been reported previously indicating that both recipients and healthcare providers tend to report preferentially the symptoms of the disease against which the non-live vaccine was administered. This bias has been linked to nocebo behavior [17] and investigators involved in vaccine safety surveillance should also consider this bias during the validation of potential associations that may be disproportional [31].

In our data, physician vaccine recipients more often reported post-vaccination AEs in the multivariable model including Q-No score, occupation, gender, and age. One can speculate that physicians may recognize and monitor symptoms easier than other HWCs. Nonetheless, nocebo effects do not appear to drive this association, since physicians were less likely to score more than 15 in Q-No in the same analysis.

Working in the COVID-19 era has exerted immense physical and emotional stress upon HCWs, which could predispose them to nocebo effects. In addition, the vicious cycle of negative expectations from treatment and resultant negative effects of treatment, i.e., nocebo effects, is fueled by negative media references [18,32], which for the case of pandemic are unprecedented in extent and intensity [33]. Therefore, one would expect that nocebo-prone individuals would have increased AE reporting post vaccination [34], which was not found. We believe that this is because nocebo-prone participants were more commonly unvaccinated, which did not allow us to highlight this association.

### 4.3. Study Strengths and Limitations

This is a cross-sectional and face-to-face survey to investigate vaccine hesitancy as an action/decision, not as an intention/perception, among HCWs in Athens, Greece, which provides novel evidence of an association between vaccine hesitancy and nocebo-prone behavior. The cross-sectional design of the survey limits causal interpretation of this association. However, in this setup, participants reported their decision for vaccine acceptance or hesitancy unbiased from factors that might have triggered insincere reporting in the context of a prospective survey. AEs were not ascertained in an objective manner employing specific criteria, but again, the study was participant-centered and focused on their own subjective perception referring to vaccine tolerance. Additional study limitations stemming from the use of Q-No to identify nocebo-prone behavior is the fact that it has been used in few studies [20,21,22], that it has only been validated in outpatients, rather than HCWs and that the evaluation was clinical, not in a double-blind, placebo-controlled setting [19]; however, no other such tool is available to identify potential nocebo behaviors. Furthermore, despite anonymity, many HCWs may have felt compelled to give the answer they considered most professionally appropriate, rather than the one that most sincerely expressed their outlook on vaccination and related AEs. Finally, although the rates of non-responders were low, it is possible that HCWs manifesting a strong tendency toward nocebo behaviors may be more likely to avoid participation in a study evaluating vaccine hesitancy and AEs. Finally, several previous studies showed association of SARS-CoV-2 vaccine hesitancy with influenza vaccine hesitancy as a marker, which was not studied in our survey.

### 4.4. Conclusions

In a cross-sectional survey conducted in five hospitals in Athens, in the spring of 2021, two out of eleven HCWs refused vaccination for SARS-CoV-2. Female and other-than-physician HCWs were more likely to deny vaccination, as well as HCWs with suspected nocebo sensitivity. Among the vaccinees, one in three experienced at least one AE, mostly local pain and flu-like symptoms. Female HCWs and physicians most often reported AEs after vaccination for SARS-CoV-2.

### 4.5. Interpretation

Our findings provide evidence that SARS-CoV2 vaccine hesitancy may be associated with nocebo-prone behavior, but further investigation is needed. All human beings experience nocebo behaviors which vary by person, and they are controlled by identified brain mechanisms located in labelled brain areas [12,35,36]. Both modifiable and non-modifiable factors synchronize nocebo effects, which should be considered in the campaign for SARS-CoV-2 vaccination. Among them, misinformation and disinformation (“infodemic”) are key elements magnifying nocebo effect that should be addressed appropriately [5]. HCWs play a key role in general population decision-making for vaccination, thus special educational care for this group is required to further improve vaccination rates worldwide.

## Figures and Tables

**Figure 1 vaccines-09-01179-f001:**
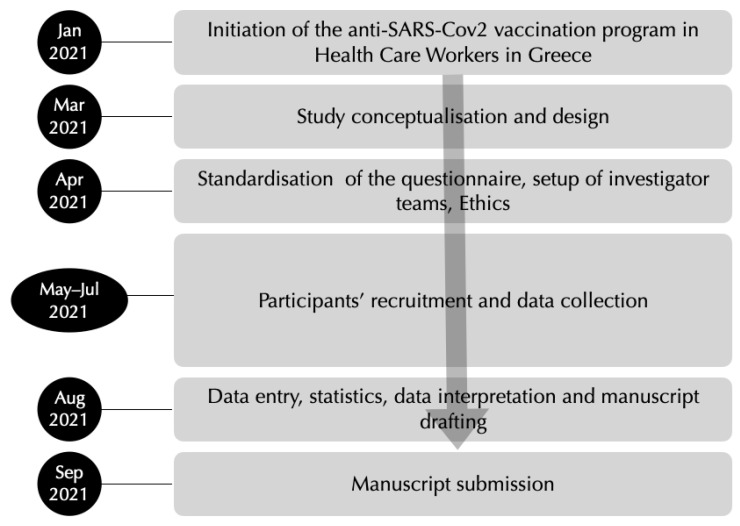
Flow diagram of the study.

**Table 1 vaccines-09-01179-t001:** The Q-No questionnaire.

Question	Rating (1 = Never, 5 = Always)
I read the summary of product characteristics (SPC) before taking a medication	
I have discontinued a medication because of adverse effects in the past	
I ask my physician for potential adverse effects of the medication he/she gives me	
I take into account the adverse effects reported in the summary of product characteristics (SPC) seriously	
total	(4–20)

**Legend:** The self-fulfilled Q-No questionnaire has been prospectively evaluated in 341 Greek out-patients after a six-month observation period for the establishment of nocebo-like behaviors. Using a cut-off at score 15, the Q-No predicts presumed nocebo susceptibility with 71.7% specificity, 67.5% sensitivity, and 42.5% positive predictive value [19]. Q-No has been used in subsequent studies contacted in Greece [20,21] and the UK [22].

**Table 2 vaccines-09-01179-t002:** Vaccination acceptance, post-vaccination adverse events and Q-No scores in the study population.

	All	Female	Male	Phys.	Nurses	Admin.	Other	Q-No ≥ 15	Q-No < 15
N	1309	866	421	557	364	148	213	325	984
%	100	67.2	32.6	43.4	28.4	11.5	16.6	24.8	75.2
Mean age	41.6	41.3	41.6	37.9	42.3	47.1	56.5	43.6	40.9
SD	10.9	10.9	11.0	10.4	10.5	9.3	10.7	11.4	10.7
Vaccinees *	1072	670	382	524	270	114	138	220	852
%	81.9	77.4	90.7	94.1	74.2	77.0	64.8	67.7	86.6
95% CI	79.7–83.9	74.5–80.0	87.6–93.2	91.8–95.8	69.4–78.4	69.5–83.1	58.1–70.9	62.4–72.6	84.3–88.6
Hesitant	237	196	39	33	94	34	75	105	132
%	18.1	22.6	9.3	5.9	25.8	23.0	25.2	32.3	13.4
95% CI	16.1–20.3	20.0–25.5	6.83–12.4	4.2–8.2	21.6–30.6	16.9–30.4	29.0–41.9	27.4–37.6	11.4–15.7
Vaccinees * with any AE	709	477	223	364	177	74	82	143	566
%	67.5	72.8	59.3	70.1	67.6	66.7	61.2	66.8	67.6
95% CI	64.6–70.2	69.3–76.1	54.3–64.2	66.0–73.9	61.6–73.0	57.4–74.9	52.7–69.1	60.2–72.8	64.4–70.7
Q-No ≥ 15	325	237	83	87	122	40	72	-	-
%	24.8	27.4	19.7	15.6	33.5	27.0	33.8	-	-
95% CI	22.6–27.2	24.5–30.4	16.8–23.8	12.8–18.9	28.9–38.5	20.5–34.8	27.7–40.4	-	-
Q-No < 15	984	629	338	470	242	108	141	-	-
%	75.2	72.6	80.3	84.4	66.5	72.9	66.2	-	-
95% CI	72.8–77.4	69.6–75.5	76.2–83.8	81.1–87.2	61.5–71.2	65.2–79.5	59.6–72.3	-	-

**Legend:** Percentages are based on the number of non-missing values. AE: adverse event; SD: standard deviation. P.: participants, Phys: physicians, Admin: administration staff; Other staff included physiotherapists, social workers, psychologists, and biologists. CI: confidence intervals. Participants with adverse events reporting at least one AE after complete vaccination (e.g., both doses of vaccines, or one dose with single-shot vaccine). (*) Vaccinees were considered participants with at least one dose of vaccine.

**Table 3 vaccines-09-01179-t003:** Multivariate logistic regression analysis for vaccine acceptance by Q-No score, gender, age, and occupation.

Vaccination	OR.	St.Err.	*p*-Value	[95% Conf	Interval]
Q-No score ≥ 15	0.425	0.073	<0.001	0.303	0.596
Age	1.013	0.008	0.091	0.998	1.029
Physicians	4.727	1.046	<0.001	3.064	7.293
Females	0.580	0.127	0.013	0.378	0.890
Constant	5.799	1.277	<0.001	3.766	8.929

**Legend:** St. Err. is the standard error of the odds ratio, approximated by the delta rule, i.e., SE(OR) = Exp(B) × SE(B), where B is the regression coefficient of the specified independent variable, and OR = Exp(B).

**Table 4 vaccines-09-01179-t004:** Severity of reported post-vaccine adverse events in vaccinee participants.

	N	%	95% CI
Vaccinees with any AE after 1st dose			
Severe AEs	4	0.4	0.1–1.0
Moderate AEs	56	5.3	4.1–6.8
Mild AEs	470	44.5	41.5–47.5
No AEs	527	49.9	46.8–52.9
Total	1057	100.00	
Missing data	15		
Vaccinees with any AE after 2nd dose			
Severe AEs	6	0.6	0.27–1.31
Moderate AEs	127	12.5	10.6–14.7
Mild AEs	437	43.1	40.1–46.2
No AEs	444	43.8	40.8–46.9
Total	1014	100.0	
Missing data	21		
Grand Total	1035 *		

**Legend:** The percentages are based on the total number of respondents without missing values. Ranking of adverse events was individual, without objective criteria. AE: adverse events. * Vaccinees with 2nd dose and vaccinees without one single-shot vaccine added.

**Table 5 vaccines-09-01179-t005:** Most common reported adverse events after 1st dose by Q-No score.

Adverse Events (AE) after 1st Dose	Overall N = 1057		Q-No Score ≥ 15 N = 217		Q-No Score < 15 N = 840		*p*-Value
	n	% (95% CI)	n	% (95% CI)	n	% (95% CI)	
At least on AE	530	50.1 47.1–53.2	106	48.8 42.2–55.5	424	50.5 47.1–53.8	0.669
Hypertension	11	1.04 0.6–1.9	1	0.4 0.7–3.2	10	1.2 0.6–2.2	0.569
Hypotension	5	0.5 0.2–1.1	1	0.4 0.01–3.2	4	0.5 0.2–1.3	>0.999
Tachycardia	25	2.4 1.6–3.5	4	1.8 0.7–4.8	21	2.5 1.6–3.8	0.751
Rash	13	1.2 0.7–2.1	4	1.84 0.7–4.8	9	1.1 0.6–2.0	0.566
Dyspnea	10	0.9 0.5–1.7	2	0.9 0.2–3.6	8	0.9 0.5–1.9	>0.999
Dizziness	53	5.0 3.8–6.5	9	4.1 2.1–7.8	44	5.2 3.9–7.0	0.630
Local pain	293	27.7 25.1–30.5	58	26.7 21.2–33.0	235	28.0 25.0–31.1	0.779
Headache	106	10.0 8.3–12.0	24	11.1 7.5–16.0	82	9.7 7.9–12.0	0.659
Fever	45	4.3 3.2–5.7	6	2.8 1.2–6.1	39	4.64 3.40–6.3	0.302
Fatigue	71	6.7 5.35–8.40	12	5.5 3.2–9.5	59	7.02 5.5–9.0	0.528
Myalgia	113	10.7 9.0–12.7	23	10.6 7.1–15.5	90	10.7 8.8–13.0	0.981

**Legend:** The percentages are based on the total number of respondents who have done both doses of vaccine and have no missing values. The categories are not mutually exclusive.

**Table 6 vaccines-09-01179-t006:** Most common reported adverse events after 2nd dose by Q-No score.

Adverse Events (AE) after 2nd Dose	Overall N = 1014		Q-No Score ≥ 15 N = 204		Q-No Score < 15 N = 810		*p*-Value
	n	% (95% CI)	n	% (95% CI)	n	% (95% CI)	
At least on AE	570	56.2 53.1–59.2	112	54.9 48.0–6.6	458	56.5 53.1–59.9	0.673
Hypertension	6	0.6 0.3–1.3	3	1.5 0.5–4.5	3	0.4 0.1–1.1	0.187
Hypotension	7	0.69 0.3–1.4	1	0.5 0.1–3.4	6	0.7 0.3–1.6	>0.999
Tachycardia	21	2.1 1.3–3.1	5	2.4 1.0–5.8	16	2.0 1.2–3.2	0.880
Rash	11	1.1 0.6–1.9	3	1.47 0.5–4.5	8	0.99 0.5–1.9	0.552
Dyspnea	10	1.0 0.5–1.8	2	1.0 0.2–3.8	8	1.0 0.5–1.9	>0.999
Dizziness	58	5.7 4.4–7.3	15	7.3 4.5–11.8	43	5.3 3.9–7.1	0.340
Local pain	234	23.1 20.6–25.8	47	23.0 17.5–29.3	187	23.1 20.3–26.1	>0.999
Headache	164	16.2 14.0–18.6	40	19.6 14.7–25.6	124	15.3 13.0–18.7	0.166
Fever	135	13.3 11.3–15.5	31	15.2 10.9–20.8	104	12.8 10.7–15.3	0.441
Fatigue	106	10.4 8.7–12.5	16	7.8 4.8–12.4	90	11.1 9.1–13.5	0.217
Myalgia	138	13.6 11.6–15.8	27	13.2 9.2–18.6	111	13.7 11.5–16.2	0.952

**Legend:** The percentages are based on the total number of respondents who have done both doses of vaccine and have no missing values. The categories are not mutually exclusive.

**Table 7 vaccines-09-01179-t007:** Multivariate logistic analysis for adverse event reporting by Q-No score, gender, age, and occupation.

Adverse Event	OR	St.Err.	*p*-Value	[95% Conf.	Interval]
Q-No score ≥ 15	0.901	0.159	0.555	0.638	1.273
Age	0.990	0.007	0.181	0.977	1.004
Physician	1.474	0.239	0.017	1.073	2.025
Female	2.001	0.312	<0.001	1.482	2.725
Constant	1.160	0.193	0.369	0.838	1.607

**Legend:** St. Err. is the standard error of the odds ratio, approximated by the delta rule, i.e., SE(OR) = Exp(B) × SE(B), where B is the regression coefficient of the specified independent variable, and OR = Exp(B).

## Data Availability

Study row data are available upon request.

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
