# Peer review of "Nocebo-Prone Behavior Associated with SARS-CoV-2 Vaccine Hesitancy in Healthcare Workers"

_vaccines, 2021, doi:10.3390/vaccines9101179_

Round 1

Reviewer 1 Report

The manuscript requires considerable editing to correct statistical errors and confusing statements. The adequacy of the previous validation of the Q-No instrument is questionable, however in my opinion this is not fatal, as long as this is made clear to readers. The practical significance of the findings is questionable. Even if the factors measured by the Q-No questionnaire are associated with vaccine hesitancy, it is difficult to see they could be modified through a communication campaign. Specific recommendations are listed below.

In the title, replace "contributes to" with "associated with". You have not shown a cause-effect relationship.

Abstract:

Rephrase the first sentence of the abstract as, "Among healthcare workers (HCWs), SARS-CoV-2 vaccine hesitancy may be linked to a higher susceptibility to nocebo effects, i.e. adverse events (AEs) experienced after medical treatments due to negative expectations."

P1, L4: Replace "included a validated tool" with "that included a tool". The adequacy of the previous validation is questionable (see below).

P1, L8-10: Replace the sentence beginning with "Odds of vaccination..." with "In a multivariate logistic regression model, with Q-No score, age, gender, and occupation as independent variables, estimated Odds Ratios (ORs) of vaccination were 0.43 (i.e., less likely, p<0.001) in participants with Q-No score ≥15 vs. Q-No score <15, 0.58 in females vs. males (p=0.013), and 4.7 (i.e., more likely, p<0.001) in physicians vs. other HCWs, independently of Age which was not significantly associated with OR of vaccination." Do not express OR as % because this confuses OR with Attributable Risk %.

P1, L10: Replace "AE" with "adverse effect (AE)".

P1, L11-12: Replace the sentence beginning with "Females and physicians..." with "In a multivariate logistic regression model, with Q-No score, age, gender, and occupation as independent variables, estimated Odds Ratios (ORs) of AE reporting were 2.0 in females vs. males (p<0.001) independently of Age, Q-No score and occupation, which were not significantly associated with OR of AE."

P1, L14: Replace "the nocebo effect" with "nocebo-prone people".

Putting research into context:

P2, L2: Replace "attributed" with "attributable".

P2, L3: After "negative expectations" insert "and not the treatment itself".

P2, L13: Replace "may be an effective way" with "are worthy of investigation as a potential way".

Introduction:

P3, L4-5: Replace "can predict the future development of nocebo effects among neurologic outpatients[19]" with "can predict a neurologist's opinion that a person is prone to nocebo effects[19]". Has there ever been a study where a placebo was administered to subjects (as in the context of a placebo-controlled clinical trial of an active therapy), frequency of adverse effects to the placebo were measured, and Q-No score was shown to be predictive of likelihood of adverse effects to the placebo? Unless this has been done, I would argue that one can't assert that the Q-No questionnaire has be shown to have predictive validity for nocebo effects.

Methods:

P3, L21: Delete the phrase "which has been validated to detect predisposition to nocebo-related symptoms."

P3, L23: Replace "for the future development of nocebo effects[19] making it a reliable instrument" with "for presumed nocebo susceptibility [19]". Sensitivity and specificity do not assess the reliability (consistency, repeatability) of a test.

Table 1:

Table1 has been incorrectly labelled as "Table 2". Correct this.

Legend: Replace "Q-No predicts nocebo" with "Q-No predicts presumed nocebo susceptibility".

Findings:

P4, L15:  Replace "421 (32·6%) were female and 866 (67·2%) males" with "421 (32·6%) were male and 866 (67·2%) females.

P5, L25-30: This paragraph is confusing. I advise rephrasing as, "In a multivariate logistic regression model (Table 3), with logit of vaccine acceptance as the dependent variable (the outcome), and Q-No score, age, gender, and occupation as the independent variables (hypothesized predictors), estimated Odds Ratios (ORs) of vaccination were 0.43 (i.e., vaccination less likely) in participants with Q-No score ≥15 compared to those with Q-No score <15 (p<0.001 for the null hypothesis),  0.58 (vaccination less likely) in females compared to males (p=0.013), and 4.7 (vaccination more likely) in physicians compared to other HCWs (p<0.001), independently of Age which was not significantly associated with OR of vaccination."

Table 3 is difficult to understand and contains internal inconsistencies that need to be corrected.

In the title, replace "Multivariate regression"  with "multivariate logistic regression".

The table needs column headings, and footnotes to explain the columns.

Column 1 is a list of the independent variables included in the model.

Column 2 is probably the estimated Odds Ratio, which I assume was calculated as OR=Exp(B), where B is the regression coefficient of the specified independent variable.

Column 3, I am not sure, but it might be the standard error of B.

Column 4 might be the P-value for the null hypothesis that OR=1, or B=0.

Columns 5 and 6 are probably meant to be the lower and upper 95% confidence limits for OR. However, they have been incorrectly calculated, because they should be calculated as Exp[B±1.96×SE], where B=Ln(OR).

P6, L1: Replace "multivariate regression" with "multivariate logistic regression".

P6, L1: Replace "vaccines" with vaccinees".

P6, L2: Replace "table 6" with "Table 7".

P6, L2-3: Replace the phrase "female and physician vaccinees" with "female vaccinees (compared to males)". You said that physicians were more likely to report adverse effects than other HCWs, but I don't think the disparity is statistically significant. I believe the calculations in Table 7 are incorrect (see below).

Table 5: Replace "Rach" with "Rash".

Table 6: Replace "Rach" with "Rash".

Table 7 is difficult to understand and contains internal inconsistencies that need to be corrected.

In the title, replace "Multivariate regression"  with "multivariate logistic regression".

The table needs column headings, and footnotes to explain the columns.

Column 1 is a list of the independent variables included in the model.

Column 2 is probably the estimated Odds Ratio, which I assume was calculated as OR=Exp(B), where B is the regression coefficient of the specified independent variable.

Column 3, I am not sure, but it might be the standard error of B.

Column 4 might be the P-value for the null hypothesis that OR=1, or B=0.

Columns 5 and 6 are probably meant to be the lower and upper 95% confidence limits for OR. However, they have been incorrectly calculated, because they should be calculated as Exp[B±1.96×SE], where B=Ln(OR).

Also, in the "Physician" row, the p-value (stated as 0.017) does not make sense. If OR=1.474 then B=Ln(1.474)=0.388. If SE=0.239 then L95CL of OR=Exp[0.388-1.96×0.239]=0.923 and U95CL of OR=Exp[0.388+1.96×0.239]=2.35. So the p-value (2-sided) must be >0.05.

Discussion

Study strengths and limitations:

Has there ever been a study where a placebo was administered to subjects (as in the context of a placebo-controlled clinical trial of an active therapy), frequency of adverse effects to the placebo were measured, and Q-No score was shown to be predictive of likelihood of adverse effects to the placebo? Unless this has been done, one can't assert that the Q-No questionnaire has be shown to have predictive validity for nocebo effects. However, one could argue that in the study reported in this manuscript, the predictive validity of Q-No score for nocebo effects is not critically important, because you are hypothesizing that the Q-No score (whatever it measures) predicts vaccine hesitancy. But this matter needs to be discussed openly with readers.

Interpretation:

In the first sentence, replace "is linked to" with "may be associated with".

In the Q-No questionnaire, items 1, 3 and 4 appear to reflect obsessive-compulsive personality traits, and item 2 refers to past medical history. It is plausible that these factors could be associated with vaccine hesitancy, however, none of these factors are modifiable. You need to explain how communication and education could address these issues in a manner that would reduce vaccine hesitancy. I would think that most clinicians have had the experience that the more one talks about adverse effects, the more obdurate the treatment-hesitant client becomes. Either provide references that demonstrate effectiveness (not just someone's opinion), or suggest further investigations.

Author Response

No

Comment

Reply

Page

1

The manuscript requires considerable editing to correct statistical errors and confusing statements. The adequacy of the previous validation of the Q-No instrument is questionable, however in my opinion this is not fatal, as long as this is made clear to readers. The practical significance of the findings is questionable. Even if the factors measured by the Q-No questionnaire are associated with vaccine hesitancy, it is difficult to see they could be modified through a communication campaign.

Thank you for taking the time to review our paper extensively and for your comments that improved the quality of the report significantly, indeed.

NA

2

In the title, replace "contributes to" with "associated with". You have not shown a cause-effect relationship.

Changed

1

3

Rephrase the first sentence of the abstract as, "Among healthcare workers (HCWs), SARS-CoV-2 vaccine hesitancy may be linked to a higher susceptibility to nocebo effects, i.e. adverse events (AEs) experienced after medical treatments due to negative expectations."

Changed

1

4

P1, L4: Replace "included a validated tool" with "that included a tool". The adequacy of the previous validation is questionable (see below).

Changed

1

5

P1, L8-10: Replace the sentence beginning with "Odds of vaccination..." with "In a multivariate logistic regression model, with Q-No score, age, gender, and occupation as independent variables, estimated Odds Ratios (ORs) of vaccination were 0.43 (i.e., less likely, p<0.001) in participants with Q-No score ≥15 vs. Q-No score <15, 0.58 in females vs. males (p=0.013), and 4.7 (i.e., more likely, p<0.001) in physicians vs. other HCWs, independently of Age which was not significantly associated with OR of vaccination." Do not express OR as % because this confuses OR with Attributable Risk %.

Changed

1

6

P1, L10: Replace "AE" with "adverse effect (AE)".

Changed

1

7

P1, L11-12: Replace the sentence beginning with "Females and physicians..." with "In a multivariate logistic regression model, with Q-No score, age, gender, and occupation as independent variables, estimated Odds Ratios (ORs) of AE reporting were 2.0 in females vs. males (p<0.001) independently of Age, Q-No score and occupation, which were not significantly associated with OR of AE."

Changed

1

8

P1, L14: Replace "the nocebo effect" with "nocebo-prone people".

Changed

1

9

P2, L2: Replace "attributed" with "attributable".

Changed

2

10

P2, L3: After "negative expectations" insert "and not the treatment itself".

Changed

2

11

P2, L13: Replace "may be an effective way" with "are worthy of investigation as a potential way".

Changed

2

12

P3, L4-5: Replace "can predict the future development of nocebo effects among neurologic outpatients[19]" with "can predict a neurologist's opinion that a person is prone to nocebo effects[19]".

Changed

3

13

Has there ever been a study where a placebo was administered to subjects (as in the context of a placebo-controlled clinical trial of an active therapy), frequency of adverse effects to the placebo were measured, and Q-No score was shown to be predictive of likelihood of adverse effects to the placebo? Unless this has been done, I would argue that one can't assert that the Q-No questionnaire has be shown to have predictive validity for nocebo effects.

Q-No was evaluated clinically. We added this additional limitation to the limitation section.“Additional study limitations stemming from the use of Q-No to identify nocebo-prone behavior is the fact that it has been used in few studies31−33, that it has only been validated in outpatients, rather than HCWs and that the evaluation was clinical, not in a double-blind, placebo-controlled setting19; “

9

14

Methods: P3, L21: Delete the phrase "which has been validated to detect predisposition to nocebo-related symptoms."

Deleted

3

15

P3, L23: Replace "for the future development of nocebo effects[19] making it a reliable instrument" with "for presumed nocebo susceptibility [19]". Sensitivity and specificity do not assess the reliability (consistency, repeatability) of a test.

Changed

3

16

Table 1: Table1 has been incorrectly labelled as "Table 2". Correct this. Legend: Replace "Q-No predicts nocebo" with "Q-No predicts presumed nocebo susceptibility".

Corrected

3

17

P5, L25-30: This paragraph is confusing. I advise rephrasing as, "In a multivariate logistic regression model (Table 3), with logit of vaccine acceptance as the dependent variable (the outcome), and Q-No score, age, gender, and occupation as the independent variables (hypothesized predictors), estimated Odds Ratios (ORs) of vaccination were 0.43 (i.e., vaccination less likely) in participants with Q-No score ≥15 compared to those with Q-No score <15 (p<0.001 for the null hypothesis),  0.58 (vaccination less likely) in females compared to males (p=0.013), and 4.7 (vaccination more likely) in physicians compared to other HCWs (p<0.001), independently of Age which was not significantly associated with OR of vaccination."

Changed

6

18

Table 3 is difficult to understand and contains internal inconsistencies that need to be corrected.In the title, replace "Multivariate regression"  with "multivariate logistic regression". The table needs column headings, and footnotes to explain the columns. Column 1 is a list of the independent variables included in the model. Column 2 is probably the estimated Odds Ratio, which I assume was calculated as OR=Exp(B), where B is the regression coefficient of the specified independent variable. Column 3, I am not sure, but it might be the standard error of B. Column 4 might be the P-value for the null hypothesis that OR=1, or B=0.

Title changed and column headings added

6

19

P6, L1: Replace "multivariate regression" with "multivariate logistic regression".

Changed

7

20

P6, L1: Replace "vaccines" with vaccinees".

Changed

7

21

P6, L2: Replace "table 6" with "Table 7".

Changed

7

22

P6, L2-3: Replace the phrase "female and physician vaccinees" with "female vaccinees (compared to males)". You said that physicians were more likely to report adverse effects than other HCWs, but I don't think the disparity is statistically significant. I believe the calculations in Table 7 are incorrect (see below).

Changed

7

23

Table 5: Replace "Rach" with "Rash".

Replaced

7

24

Table 6: Replace "Rach" with "Rash".

Replaced

7

25

Table 7 is difficult to understand and contains internal inconsistencies that need to be corrected.In the title, replace "Multivariate regression"  with "multivariate logistic regression".The table needs column headings, and footnotes to explain the columns. Column 1 is a list of the independent variables included in the model. Column 2 is probably the estimated Odds Ratio, which I assume was calculated as OR=Exp(B), where B is the regression coefficient of the specified independent variable. Column 3, I am not sure, but it might be the standard error of B. Column 4 might be the P-value for the null hypothesis that OR=1, or B=0. Columns 5 and 6 are probably meant to be the lower and upper 95% confidence limits for OR. However, they have been incorrectly calculated, because they should be calculated as Exp[B±1.96×SE], where B=Ln(OR). Also, in the "Physician" row, the p-value (stated as 0.017) does not make sense. If OR=1.474 then B=Ln(1.474)=0.388. If SE=0.239 then L95CL of OR=Exp[0.388-1.96×0.239]=0.923 and U95CL of OR=Exp[0.388+1.96×0.239]=2.35. So the p-value (2-sided) must be >0.05.

Multivariate regression"  replaced with "multivariate logistic regression and column headings added.

 Statistics:  The St. Error Column in tables presented in the manuscript, is not the St. Error of the log (OR) so the corresponding 95% Cis cannot be calculated based on the simple formula as suggested by the reviewer.Exponentiation of the St.  Error is not appropriately as it is for the regression coefficients and the 95% CI (which is what we did). Someone cannot derive the 95% CI from provided St. Err and ORs;

The St. Err provided in the tables, is the St. Error of the OR  (not the LogOR) after been estimated using delta asymptotical approximation . Further details: in https://www.stata.com/support/faqs/statistics/delta-rule/. The naïve formula cannot be applied in approximated estimates which (for example see the 95%CI are not symmetrical)

8

26

Study strengths and limitations: Has there ever been a study where a placebo was administered to subjects (as in the context of a placebo-controlled clinical trial of an active therapy), frequency of adverse effects to the placebo were measured, and Q-No score was shown to be predictive of likelihood of adverse effects to the placebo? Unless this has been done, one can't assert that the Q-No questionnaire has be shown to have predictive validity for nocebo effects. However, one could argue that in the study reported in this manuscript, the predictive validity of Q-No score for nocebo effects is not critically important, because you are hypothesizing that the Q-No score (whatever it measures) predicts vaccine hesitancy. But this matter needs to be discussed openly with readers.

We added: “that it has only been validated in outpatients, rather than HCWs and that the evaluation was clinical, not in a double-blind, placebo-controlled settin19”. See also reply to comments No12-16. All together, make more than clear the limitations of Q-No, and protect the reader from misinterpretations.

10

27

Interpretation: In the first sentence, replace "is linked to" with "may be associated with".

Changed

10

28

In the Q-No questionnaire, items 1, 3 and 4 appear to reflect obsessive-compulsive personality traits, and item 2 refers to past medical history. It is plausible that these factors could be associated with vaccine hesitancy, however, none of these factors are modifiable. You need to explain how communication and education could address these issues in a manner that would reduce vaccine hesitancy. I would think that most clinicians have had the experience that the more one talks about adverse effects, the more obdurate the treatment-hesitant client becomes. Either provide references that demonstrate effectiveness (not just someone's opinion), or suggest further investigation.

Q-Νο should be used in a uniform manner and not in a piecemeal way, because that is how it was evaluated. The evaluation was clinical, that's true. It is also true that experimental verification would greatly strengthen the reliability of the tool. However, no other instrument there is. We suggest the campaign focusing on modifiable factors that may control nocebo, since we stated in the beginning of the paragraph that vaccine hesitancy may be associatedwith nocebo, not to aspects that the reviewer assume that are related to Q-No. We added that further investigation is needed to establish the relation between vaccine hesitancy and nocebo, of course.

10

Reviewer 2 Report

The manuscript is well written and can be consider for the publication after minor changes.

  1. Authors should add a flow diagram navigating the methods opted in this manuscript.
  2. author should add a paragraph on conclusion separately.

Author Response

No

Comment

Reply

Page

1

The manuscript is well written and can be consider for the publication after minor changes.

Thank you for taking the time to review our paper and for your comments

NA

2

Authors should add a flow diagram navigating the methods opted in this manuscript.

Figure 1 added (see below)

4

3

Αuthor should add a paragraph on conclusion separately.

Added  “In a cross-sectional survey conducted in five Hospitals in Athens, in the spring of 2021, two out of 11 HCWs refused vaccination for SARS-CoV-2. Female and non-physicians HCWs were more likely to refuse vaccination, as well as HCWs with suspected nocebo sensitivity. Among the vaccinees, one in three experienced at least one AE, mostly local pain and flu-like symptoms. Female HCWs and physicians most often reported AEs after vaccination for SARS-CoV-2.”

10

Reviewer 3 Report

Dear Authors, I want to thank you for writing the paper "NOCEBO-PRONE BEHAVIOR CONTRIBUTES TO SARS-CoV-2 VACCINE HESITANCY IN HEALTHCARE WORKERS".

I find the article innovative, and the conclusions may direct the benefit of a campaign focused on the nocebo-effect.

In Table 5, I believe "Rach" must be changed to "Rash".

 I have no other suggestion for Authors. 

Author Response

No

Comment

Reply

Page

1

Dear Authors, I want to thank you for writing the paper "NOCEBO-PRONE BEHAVIOR CONTRIBUTES TO SARS-CoV-2 VACCINE HESITANCY IN HEALTHCARE WORKERS".

We thank the reviewer for taking the time to review our paper and for the comments

NA

2

I find the article innovative, and the conclusions may direct the benefit of a campaig focused on the nocebo-effect.

Thank you for your comments

NA

3

In Table 5, I believe "Rach" must be changed to "Rash".

Correct

7

4

 I have no other suggestion for Authors

Thank you

NA

Round 2

Reviewer 1 Report

Tables 3 and 7:

I understand your explanation of the "Standard Error" and I have verified that your 95% confidence limits around the estimated Odds Ratios are correct, calculated as  Exp[Ln(OR)±1.96×SE(OR)/OR]. But other readers may have the same misunderstanding as I did initially, so you need to put a footnote below Table 3 and also below Table 7 explaining that, "St. Err. is the Standard Error of the Odds Ratio, approximated by the delta rule, i.e., SE(OR)=Exp(B)×SE(B), where B is the regression coefficient of the specified independent variable, and OR=Exp(B)."

Author Response

We added this explanation to table 3 and 7 legend as the reviewer suggested.